# Trends and determinants of caesarean section in South Asian countries: Bangladesh, Nepal, and Pakistan

Md Sohel Rana[1]*, Shrabanti Mazumder[2], Md Tareq Ferdous Khan[3], Md Mobarak Hossain Khan[4], Md Mijanur Rahman[5]

1 Department of Statistics, Comilla University, Cumilla, Bangladesh, 2 Division of Biostatistics and Bioinformatics, Department of Environmental and Public Health Sciences, College of Medicine, University of Cincinnati, Cincinnati, OH, United States of America, 3 Department of Public Health Sciences, Clemson University, Clemson, SC, United States of America, 4 Department of Social Relations, East West University, Aftabnagar, Dhaka, Bangladesh, 5 The Daffodil Centre, A Joint Venture with Cancer Council NSW, The University of Sydney, Sydney, NSW, Australia

* sohel.573@gmail.com

## Abstract

### Background

The prevalence of caesarean sections (C-sections) has remarkably increased in the past few decades worldwide, especially in the lower and middle-income countries (LMICs). To our best knowledge, no studies focused on and compared the C-section scenarios of Bangladesh, Nepal, and Pakistan based on the latest demographic and health survey (DHS) data.

### Objectives

To assess the trends and factors associated with C-sections in the three South Asian countries.

### Study population

Mothers aged 15–49 years participated in DHS 1990 to 2017–2018 and gave birth within three years of each of the surveys in Bangladesh, Nepal, and Pakistan.

### Materials and methods

This study analyzed data from five recent DHS rounds in Bangladesh and four in Nepal and Pakistan. Multivariable logistic regression was used to assess the association between C-sections and sociodemographic characteristics.

### Results

The results show that institutional delivery and C-sections have increased throughout the period in all three countries. In Bangladesh, the hospital birth rate increased from 10.0% in 2004 to 49.9% in 2017, and the corresponding figures [S1 Appendix: Figure A1 and

**Data Availability Statement:** All data are freely available on DHS (https://www.dhsprogram.com/Data/).

**Funding:** The authors received no specific funding for this work.

**Competing interests:** The authors have declared that no competing interests exist.

Figure A2] for C-sections increased from 3.5% to 32.8%. In Nepal, the hospital birth rate increased from 11.0% in 2001 to 58.6% in 2016, and the C-sections from 0.8% to 11.0%. Pakistan observed a sharp increase from 13.7% to 66.3% and 2.7% to 22.3% in the respective cases from 1990 to 2017. Results from regression reveal that the mother's age, place of residence, education, partner's education, wealth status, birth order, number of antenatal care visits, and body mass index are associated with C-section deliveries in all three countries.

## Conclusions

Our findings regarding the association of sociodemographic factors with increased C-sections may help identify subgroups of women susceptible to C-sections and offer better support regarding C-sections plans. However, the substantial increase in C-sections across the three countries warrants further investigation to identify the reasons.

## 1. Introduction

Clinically when normal vaginal delivery of a baby is considered risky or life-threatening for both mother and baby or any of them, the C-section is a preferable method. In this method, delivery is carried out through scarification of the abdomen and uterus. Despite its value of safety, C-section also holds complications [1]. Due to substantially increasing C-section rates in the LMICs, it is only recommended to practice when there are no other alternatives and when it is medically justifiable [2]. During the past 30 years, there was an increasing trend of practice of C-section; it varies from 5% to 43% among countries around the world. The recent study, which included data from 154 countries, reported an overall C-section rate of 21.1% [3]. Previously, it was observed that economically developed countries have a higher rate of C-sections [3, 4]. However, the scenarios have changed dynamically and observed a growing rate of C-sections in economically less developed countries. The developed countries are strictly imposing restrictions on unnecessary C-sections [5]. The rate of C-sections has increased globally across all regions since 1990. The sub-regions that experienced the largest increases were Eastern Asia, Western Asia, and Northern Africa, with respective increases of 44.9, 34.7, and 31.5 percentage points. On the other hand, sub-Saharan Africa and Northern America had the lowest increases, with 3.6 and 9.5 percentage points, respectively [3].

The reasons for increased use of C-section is attributed to the reduction in mortality risk of mother and child during delivery [3, 6, 7]. Since C-section is a major abdominal surgery, elective C-section should be avoided. It may cause infection, injury, and short or long-term disability for both mother and baby. Sometimes, it may interrupt the start of breastfeeding, and it may also require hysterectomy [8]. C-sections may result in permanent pain and cause complications in future deliveries [9, 10]. Moreover, a positive association between C-section and respiratory diseases and other pulmonary infections is evident [11, 12]. South Asian countries are economically, socially and politically diversified. They attempt to minimize regional health challenges [13], encourage deliveries under the supervision of skilled birth attendants [14], and maintain hygiene. C-sections has been doubled during the period 2000–2015 in the South Asian region, and the yearly increase rate is more than 5% [4]. In 2015, the C-section rate in this region was 18.1%, which exceeded the world Health Organization (WHO) recommended range 10–15% [15]. Although there is a controversy and there is no international consensus

regarding this rate, we still kept this number for baseline comparison [16]. So, it is clear that this life-saving method of delivery is being used at a higher than expected, irrespective of the medical justification in many South Asian countries [4].

According to the WHO Report 2010, there were 6 million elective C-sections practiced that cost US$ 2.32 billion [15]. Along with other causes, increasing the income of healthcare providers is a significant cause of increasing C-sections [17]. Elective C-section causes adverse impacts on health as well as an economic burden as they usually require to stay in the hospital longer [18–20].

The association between C-section and several other factors, including maternal age, maternal weight, number of times an individual woman has given birth (parity), prolonged labour, HIV infection, previous C-section delivery, dystocia, breech presentation, placenta previa, and suspected fatal complications are well documented in the literature [21, 22]. The place of residence, educational status [23], and choice of place of delivery [24] are also found to be associated. Several other significantly associated factors include baby size, higher number of antenatal care (ANC) visits, antenatal care (ANC) from private institutions, working husband, first birth and delivery in private hospitals [22, 23, 25]. A contemporary Lancet publication showed a significant difference in the use of C-sections among women in the two extreme wealth quintiles in 82 LMICs [4].

Although some studies dealt with the prevalence and determinants of C-sections in Bangladesh and South Asia [24–28], to the best our knowledge no studies have focused on the most recent scenarios of the trends and determinants of increased use of C-sections across three neighbouring countries, including Bangladesh, Nepal, and Pakistan. These three countries face similar challenges in maternal and child health issues [29, 30]. Despite some differences in specific indicators and methods, they possess similarities in terms of maternal and child health: high maternal and child mortality rates, limited access to quality health care, malnutrition, maternal health services, and socio-economic and cultural factors [31, 32]. Moreover, all three countries face challenges regarding healthcare infrastructure, skilled healthcare professionals, and availability of essential drugs and equipment. Thus, this study focused on these three South Asian countries to determine and compare (i) the trends in the use of the C-section over time and (ii) sociodemographic factors associated with the use of the C-section. It would facilitate understanding the patterns and identifying the socioeconomic and sociodemographic vulnerable cohorts with increased use of the C-section.

## 2. Materials and methods

### 2.1 Data source and study design

This study used demographic and health survey datasets from Bangladesh, Nepal and Pakistan. Datasets were extracted from the child recode file. For Bangladesh, the last five rounds of demographic and health survey (DHS) from 2004 to 2017–2018; for Nepal, the recent four rounds of DHS from 2001 to 2016, and for Pakistan available four rounds of DHS from 1990–1991 to 2017–2018 were used in analyses. The details of study design, questionnaires, and sampling methods were discussed elsewhere [33–35]. We briefly added the procedures for Bangladesh here, and almost similar procedures were followed for Nepal and Pakistan. As a part of the Demographic and Health Survey (DHS), Bangladesh DHS (BDHS) is usually conducted in every three years. The BDHS takes place in seven administrative divisions: Dhaka, Chittagong, Rajshahi, Khulna, Rangpur, Barisal, and Sylhet across both urban and rural areas. The sampling technique applied for each round of BDHSs was two-stage stratified cluster random sampling. National Population and Housing Census (NPHC) provided the enumeration areas (EAs) to determine the sampling frame of surveys. Bangladesh Bureau of Statistics (BBS)

conducts NPHC every ten years. In 2011, 2014, and 2017–18 BDHS, in the first stage, 600 enumeration areas (EAs) from 296,718 EAs, each with 120 households on an average, from 2011 NPHC were chosen with probability proportional to each unit size. Where 393 EAs were from rural areas and 207 EAs from urban. Hereafter, in the second stage, 30 households from each EA were selected by applying the systematic random sampling procedure.

In 2004 and 2007 BDHSs, the same processes were used to choose 361 EAs (227 from rural and 134 from urban) from 259,532 EAs of 2001 NPHC. Every wave had an overall 98% response rate. The National Research Ethics Committee of Bangladesh has reviewed and approved the survey protocol. The institutional review board of ORC Macro (Macro International Inc.) also approved the data collection procedures. Before data collection, informed consent was ensured from every participant. Generally, BDHSs were administered by the National Institute for Population Research and Training (NIPORT), Bangladesh, in partnership with USAID and ICF International (Fairfax, VA, USA). The ICF International provides authorization to access the data for valid research purposes. We obtained authorization to use the relevant de-identified data from the DHS authority.

BDHS usually collects data using four questionnaires: a household questionnaire, a women's questionnaire, a men's questionnaire, and a community questionnaire. For Bangladesh, a Technical Working Group (TWG) decides on the contents of the questionnaires and adopts them from the model DHS questionnaire. Data were collected on several indicators, including maternal healthcare, mortality, fertility, family planning, and nutrition. Trained staffs were recruited to collect the data, and skilled supervisors were employed to supervise the data collection process. The fieldworks were also supervised by representatives from USAID, ICF International, and NIPORT. A research organization, "Mitra and Associates," conducted these surveys. Face-to-face interview techniques was used in all data collection procedures of BDHSs.

## 2.2 Outcome variable

The outcome of interest is C-section delivery, a binary response variable with options "Yes" and "No." Only institutional deliveries, consisting of both vaginal delivery and C-section, held in governmental or non-governmental healthcare centres were considered for eligible responses.

## 2.3 Explanatory variables

Considering existing literature and availability of data from DHSs, the following sociodemographic and economic predictors were considered as explanatory variables in our study: age of mothers, educational status (no education, primary, secondary, tertiary), wealth quintile (poorest, poorer, middle, richer, richest), place of residence (urban, rural), spouse education (no education, primary, secondary, tertiary), parity, number of antenatal care (ANC) visits, place of delivery (home, hospital), Place of birth (private and public hospitals), body mass index (BMI) (underweight, normal weight, overweight, obese).

## 2.4 Statistical analyses

Univariate analyses were performed to demonstrate the characteristics of the study participants. We used Cochran-Armitage test to check the trends. Bivariate analyses (Chi-squared tests) were also carried out to assess the association between the outcome variable and the explanatory variables. The multivariable logistic regression model was fitted to assess the association between each of the socioeconomic and sociodemographic factors after adjusting the effects for others. We performed the data analyses (chi-squared tests and multivariable logistic

regression) on the pooled data set across the countries. For model selection, the backward elimination procedure was followed. Proper weighting methods were applied since all the DHSs were conducted using complex sampling techniques. We used statistical software STATA (Stata Statistical Software: Release 14, Stata Corp., College Station, TX) [36] and R (version 3.6.0) [37] to analyse the data. For statistical significance in all our analyses, we considered the two-sided p-value less than or equal to 0.05.

## 3. Results

Table 1 presents the places of deliveries for different DHS rounds in three South Asian countries. The number of deliveries at hospitals in each country has increased over time (S1 Appendix: Figure A1). In 2004, the prevalence of delivery at hospitals in Bangladesh was only 10.0%, which increased almost fivefold (49.9%) in 2017. In Nepal, this prevalence was 11.0% in 2001, which also increased about five times (58.6%) in 2016. The scenario was similar for Pakistan (13.7% in 1990 and 66.3% in 2017).

Table 2 summarizes the distribution of C-section deliveries by country and survey years. We also demonstrated the trend of C-section deliveries in [S1 Appendix: Figure A2]. In 2004, the C-section rate in Bangladesh was about 3.5%, which increased to 32.8% in 2017. Nepal had relatively the lowest C-section rate among the three countries. In 2001, the rate was 0.8%, while 2016 it increased to 9.0%. Pakistan also demonstrated a sharp increase from 1990 to 2017 (2.7% to 22.3%).

Table 3 shows that the C-section rate in urban areas was higher compared to rural areas. In Bangladesh and Pakistan, the urban C-section rates were approximately two times higher (27.2% vs 12.0% for Bangladesh and 27.8% vs 13.9% for Pakistan) than the rural rates. In Nepal, the C-section rate in urban areas was 10.0%, whereas it was only 2.0% in rural areas. The likelihood of the C-section among mothers with higher education was higher, and a

**Table 1. Place of birth reported at different demographic health survey rounds across three countries: Bangladesh, Nepal, and Pakistan.**

| Country | Survey year | Total births | Place of birth | |
|---|---|---|---|---|
| | | | At home | At hospital |
| | | | % (n) | % (n) |
| **Bangladesh** | 2004 | 6986 | 90.1 (6291) | 10.0 (695) |
| | 2007 | 6055 | 85.0 (5148) | 15.0 (907) |
| | 2011 | 8777 | 75.0 (6584) | 25.0 (2193) |
| | 2014 | 4625 | 61.3 (2836) | 38.7 (1789) |
| | 2017–18 | 5331 | 50.1 (2670) | 49.9 (2661) |
| | P-value | | P<0.01 | |
| **Nepal** | 2001 | 6971 | 89.0 (6202) | 11.0 (769) |
| | 2006 | 5545 | 81.0 (4492) | 19.0 (1053) |
| | 2011 | 5391 | 63.1 (3402) | 36.9 (1990) |
| | 2016 | 5060 | 41.4 (2096) | 58.6 (2964) |
| | P-value | | P<0.01 | |
| **Pakistan** | 1990 | 6368 | 86.4 (5499) | 13.7 (869) |
| | 2006 | 9110 | 64.8 (5901) | 35.2 (3209) |
| | 2012 | 11955 | 51.7 (6176) | 48.3 (5779) |
| | 2017 | 10479 | 33.7 (3533) | 66.3 (6946) |
| | P-value | | P<0.01 | |

Note: % are weighted to account for survey design. P-values were obtained from the Cochran-Armitage test.

**Table 2. Delivery by C-section in Bangladesh, Nepal and Pakistan at different DHS rounds.**

| Country | Survey year | Total births | Caesarean section | |
|---|---|---|---|---|
| | | | **No** | **Yes** |
| | | | **% (n)** | **% (n)** |
| **Bangladesh** | 2004 | 6987 | 96.5 (6745) | 3.5 (242) |
| | 2007 | 6055 | 92.5 (5599) | 7.5 (457) |
| | 2011 | 8777 | 85.9 (7542) | 14.1 (1235) |
| | 2014 | 4626 | 75.7 (3504) | 24.3 (1122) |
| | 2017–18 | 5331 | 67.2 (3584) | 32.8 (1747) |
| | P-value | | P<0.01 | |
| **Nepal** | 2001 | 6977 | 99.2 (6919) | 0.8 (58) |
| | 2006 | 5545 | 97.3 (5397) | 2.7 (148) |
| | 2011 | 5391 | 95.4 (5143) | 4.6 (248) |
| | 2016 | 5060 | 91.0 (4603) | 9.0 (457) |
| | P-value | | P<0.01 | |
| **Pakistan** | 1990 | 6373 | 97.3 (6199) | 2.7 (175) |
| | 2006 | 9110 | 92.7 (8444) | 7.3 (665) |
| | 2012 | 11955 | 85.9 (10270) | 14.1 (1685) |
| | 2017 | 10482 | 77.7 (8143) | 22.3 (2339) |
| | P-value | | P<0.01 | |

Note: % are weighted to account for survey design. P-values were obtained from the Cochran-Armitage test.

similar scenario was evident for husbands with higher education. Among women without any formal education, the C-section rates were 3.1%, 1.1%, and 8.6% for Bangladesh, Nepal and Pakistan, respectively. In contrast, among women with more than twelve years of education, the C-section rates were 59.2%, 23.8%, and 50.4% for Bangladesh, Nepal and Pakistan, respectively. A significant difference in the C-section rates was also evident in the economic status of women. In Bangladesh, only 2.9% received C-section delivery among those in the poorest quantile, whereas 37.7% in the richest quantile. A similar pattern was also observed for Nepal (0.4% vs 11.2%) and Pakistan (8.5% vs 45.5%). The birth order observed a distributional difference in the C-section rates. The highest rates were evident for the first birth in all countries (Bangladesh at 25.5%, Nepal at 8.7%, and Pakistan at 28.3%). Bangladeshi women seeking more than four antenatal care visits had the highest rate of C-sections at 40.1%, Pakistani was in second at 36.0%, and in Nepal, the rate was 13.3%. However, with no antenatal visits, the prevalence of C-sections was low at 2.6%, 0.5%, and 2.4%, respectively for Bangladesh, Nepal and Pakistan. The C-section rate among Bangladeshi obese women was the highest (55.2%), followed by Pakistan at 39.0%, and the lowest was in Nepal at 19.7%.

Table 4 presents the adjusted odds ratio (crude odds ratio enclosed in S1 Table in S1 Appendix) and the corresponding 95% confidence interval. Age was significantly associated with increasing likelihood of C-section (Bangladesh: AOR = 1.05, CI = 1.03, 1.07; Nepal: AOR = 1.08, CI = 1.06, 1.11; Pakistan: AOR = 1.02, CI = 1.00, 1.04). A significant association with place of residence was only observed in Nepal (AOR = 1.20, CI = 1.05, 1.50). A tendency for a higher likelihood of C-sections among educated women was observed both in Bangladesh and Pakistan compared to those with no formal education. The AOR was 2.45 (CI = 1.88, 3.35) for women with more than 12 years of education in Bangladesh, while it was 2.34 (CI = 1.70, 3.24) in Pakistan. A higher likelihood was also evident for the first birth in all countries. Women with more than four antennal care visits showed a higher likelihood of C-sections in all three countries (Bangladesh: AOR = 7.03, CI = 5.56, 8.66; Nepal: AOR = 5.27,

**Table 3. Association between C-section delivery and socio-demographic factors in Bangladesh, Nepal, and Pakistan.**

| Variable | Bangladesh | Nepal | Pakistan |
|---|---|---|---|
| | % (95% CI) | % (95% CI) | % (95% CI) |
| **Current Age** | | | |
| 15–19 Years | 14.5 (13.2–15.9) | 3.6 (2.6–4.9) | 13.4 (8–21.4) |
| 20–29 Years | 15.9 (15.1–16.8) | 3.3 (3–3.7) | 19.0 (16.4–21.8) |
| 30–39 Years | 15.9 (14.5–17.4) | 3.1 (2.6–3.7) | 19.1 (16.4–22.2) |
| 40–49 Years | 6.9 (5–9.4) | 2.4 (1.5–3.7) | 8.7 (5.7–13.1) |
| | P<0.001 | | |
| **Place of residence** | | | |
| Urban | 27.2 (25.5–29) | 10.0 (8.9–11.3) | 27.8 (24.7–31.2) |
| Rural | 12.0 (11.2–12.7) | 2.0 (1.8–2.2) | 13.9 (11.6–16.6) |
| | P<0.001 | | |
| **Women's education** | | | |
| No formal education | 3.1 (2.6–3.6) | 1.1 (1.0–1.4) | 8.6 (7.0–10.5) |
| 1–5 years | 8.1 (7.4–9.0) | 3.1 (2.5–3.8) | 16.7 (13.3–20.7) |
| 6–12 years | 24.7 (23.6–25.8) | 8.6 (7.8–9.6) | 31.5 (27.5–35.8) |
| More than 12 years | 59.2 (54.9–63.3) | 23.8 (13.1–39.2) | 50.4 (41.5–59.2) |
| | P<0.001 | | |
| **Wealth index (1–5)** | | | |
| Poorest | 2.9 (2.3–3.6) | 0.4 (0.2–0.6) | 8.5 (6.3–11.3) |
| Poorer | 6.7 (5.9–7.5) | 2.6 (2.1–3.2) | 10.8 (8.6–13.4) |
| Middle | 12.9 (11.8–14) | 1.3 (0.9–1.8) | 20.3 (16.9–24.3) |
| Richer | 23.7 (22.3–25.2) | 2.8 (2.3–3.5) | 31.9 (28.2–35.9) |
| Richest | 37.7 (35.8–39.7) | 11.2 (10–12.4) | 45.5 (41.2–49.9) |
| | P<0.001 | | |
| **Spouse Education** | | | |
| No formal education | 4.7 (4.1–5.3) | 1.0 (0.7–1.3) | 9.6 (7.6–12.1) |
| 1–5 years | 10.5 (9.6–11.4) | 1.7 (1.3–2.2) | 11.1 (8.3–14.7) |
| 6–12 years | 23.6 (22.5–24.9) | 5.2 (4.7–5.7) | 24.4 (21.1–28) |
| More than 12 years | 48.8 (45.8–51.7) | 13.4 (9.9–17.9) | 35.3 (28.4–43) |
| | P<0.001 | | |
| **Birth order** | | | |
| 1 | 25.5 (24.2–26.8) | 8.7 (7.8–9.8) | 28.3 (24.0–33.0) |
| 2 | 17.1 (16.1–18.2) | 3.9 (3.4–4.5) | 25.0 (21.3–29.2) |
| 3 | 11.3 (10.2–12.5) | 1.5 (1.1–2.0) | 23.8 (19.2–29.1) |
| 4 | 5.7 (4.7–6.8) | 0.5 (0.3–1.0) | 15.6 (11.3–21.1) |
| 5 or more | 2.3 (1.7–2.9) | 0.6 (0.4–1.0) | 8.2 (6.2–10.6) |
| | P<0.001 | | |
| **Number of ANC visits** | | | |
| 0 | 2.6 (2.2–3.1) | 0.5 (0.3–0.8) | 2.4 (1.3–4.2) |
| 1 | 8.5 (7.4–9.6) | 0.3 (0.1–1.0) | 7.9 (4.8–12.8) |
| 2 | 16.6 (15.2–18.1) | 1.0 (0.6–1.7) | 12.1 (8.6–16.9) |
| 3 | 19.4 (17.8–21.1) | 2.5 (1.9–3.3) | 19.1 (14.9–24.1) |
| 4 | 28 (25.8–30.3) | 4.9 (4.0–6.1) | 23.8 (18.9–29.6) |
| More than 4 | 40.1 (38.4–41.9) | 13.3 (12.0–14.8) | 36.0 (32.6–39.6) |
| | P<0.001 | | |
| **BMI** | | | |
| Underweight | 6.9 (6.2–7.7) | 1.4 (1.1–1.8) | 11.0 (7.8–15.2) |

(Continued)

**Table 3.** (Continued)

| Variable | Bangladesh | Nepal | Pakistan |
|---|---|---|---|
| | **% (95% CI)** | **% (95% CI)** | **% (95% CI)** |
| Normal weight | 14.5 (13.8–15.3) | 2.6 (2.3–2.9) | 12.4 (10.5–14.5) |
| Overweight | 37.9 (35.7–40.2) | 16.1 (13.9–18.5) | 23.4 (19.9–27.2) |
| Obese | 55.2 (50.2–60.1) | 19.7 (13.6–27.6) | 39.0 (32.8–45.6) |
| | P<0.001 | | |
| Type of hospital | | | |
| Public | 36.2 (34.6–37.9) | 10.9 (10.0–11.8) | 24.4 (23.3–25.5) |
| Private | 77.5 (76.2–78.6) | 27.4 (25.0–29.9) | 31.3 (30.5–32.2) |
| | P<0.001 | | |

Note: % are weighted to account for survey design.

CI = 4.09, 9.15; Pakistan: AOR = 5.01, CI = 4.33, 8.90). A significantly positive association between the C-section and body mass index was observed. In Bangladesh, Nepal, and Pakistan, the AORs of C-sections for obese women were 4.25 (CI = 3.29, 5.51), 2.51 (CI = 1.33, 3.67), and 2.11 (CI = 1.51, 2.93) compared to underweight women. The significantly increasing odds of C-sections in private hospitals compared to the public hospitals were observed in all three countries (Bangladesh: AOR = 5.31, CI = 4.76, 5.93; Nepal: AOR = 2.03, CI = 1.61, 2.56); Pakistan: AOR = 1.30, CI = 1.08, 1.54).

## 4. Discussion

In this study, we investigated the trends and prevalence of C-sections across three neighbouring South Asian countries and assessed the association of sociodemographic factors with the uptake/prevalence of C-sections. We observed that there were dramatic increases in the number of deliveries at hospitals as well as C-sections across three countries over time. While this finding indicates a noticeable improvement in healthcare facilities and health service provision, the rapid increase in C-sections causes an huge economic burden on the healthcare system and may have negative health effects for both mother and kids if coupled with unnecessary C-sections [38]. The prevalence of C-sections in Bangladesh and Pakistan was almost consistent with other LMICs [39–41]. However, both countries exceeded the WHO recommended standard level of C-sections [15]. Although there is a controversy and there is no international consensus regarding this rate, we used it for baseline comparison [16]. While the prevalence of C-sections dramatically increased in Nepal over the 16 years, the rate was well below the WHO recommended range in compared to other countries. The prevalence of C-sections in other Asian countries, for example in India and Indonesia, also increased over a similar period [42, 43]. While the improvement of healthcare facilities may partly contribute to the increased prevalence of C-sections, unnecessary C-sections also occur due to mothers' misconceptions of delivery pain, convenience, and lack of patience [22]. Besides, relative underpayment of physicians across the three countries, many intend to have inappropriate financial benefits, which may also result an upsurge in elective C-sections [44].

We found that several factors were associated with the increased prevalence of C-sections across the three South-Asian countries. Consistent with other studies [45, 46], increased age of pregnancy/delivery was associated with a higher likelihood of receiving a C-section. Women who had a pregnancy at a relatively higher age underwent high risks of preterm delivery, low birth weight, perinatal death, and C-section [47–49]. We also observed a significant difference

**Table 4. Adjusted odds ratio (AOR) and 95% confidence interval (CI) for the risk of C-section corresponding to the associated factors of Bangladesh, Nepal and Pakistan.**

| Variable | Bangladesh | Nepal | Pakistan |
|---|---|---|---|
| | AOR (95% CI) | AOR (95% CI) | AOR (95% CI) |
| **Current Age** | 1.05 (1.03–1.07)*** | 1.08 (1.06–1.11)*** | 1.02 (1.00–1.04)** |
| **Place of residence** | | | |
| Rural[R] | 1 | 1 | 1 |
| **Urban** | 1.03 (0.95–1.20) | 1.20 (1.05–1.50)** | 0.91 (0.76–1.10) |
| **Women's education education** | | | |
| No formal education[R] | 1 | 1 | 1 |
| **1–5 years** | 1.30 (1.08–1.59)** | 1.03 (0.75–1.50) | 1.42 (0.99–1.84) |
| **6–12 years** | 1.82 (1.63–2.42)*** | 1.01 (0.76–1.41) | 1.43 (1.13–1.81)** |
| **More than 12 years** | 2.45 (1.88–3.35)*** | 1.01 (0.4–2.51) | 2.34 (1.70–3.24)*** |
| **Wealth index (1–5)** | | | |
| Poorest[R] | 1 | 1 | 1 |
| **Poorer** | 1.41 (1.18–1.92)*** | 2.24 (0.95–4.57) | 0.83 (0.67–1.15) |
| **Middle** | 1.92 (1.49–2.65)*** | 1.39 (0.73–2.64) | 1.39 (0.79–2.7) |
| **Richer** | 2.72 (2.23–3.71)*** | 1.57 (1.04–3.48)* | 1.91 (1.09–3.32)** |
| **Richest** | 3.33 (2.68–4.50)*** | 3.86 (1.87–6.04)* | 1.95 (0.97–3.56)* |
| **Spouse education** | | | |
| No formal education[R] | 1 | 1 | 1 |
| **1–5 years** | 1.15 (0.91–1.37) | 1.14 (0.77–1.87) | 0.64 (0.48–0.90)** |
| **6–12 years** | 1.46 (1.17–1.68)*** | 1.20 (0.81–1.89) | 0.92(0.72–1.16) |
| **More than 12 years** | 1.93 (1.45–2.20)*** | 1.63 (0.88–2.99)* | 1.01 (0.63–1.20) |
| **Birth order** | | | |
| 1[R] | 1 | 1 | 1 |
| **2** | 0.50 (0.46–0.59)*** | 0.52 (0.41–0.67)*** | 0.87 (0.69–1.11) |
| **3** | 0.33 (0.31–0.41)*** | 0.35 (0.23–0.52)*** | 1.01 (0.79–1.32) |
| **4** | 0.25 (0.16–0.30)*** | 0.16 (0.08–0.34)*** | 0.51 (0.37–0.70)*** |
| **5 or more** | 0.10 (0.07–0.14)*** | 0.22 (0.12–0.42)*** | 0.45 (0.32–0.63)*** |
| **Number of ANC visits** | | | |
| 0[R] | 1 | 1 | 1 |
| **1** | 1.97 (1.70–2.67)*** | 0.47 (0.13–1.79) | 2.13 (1.20–3.83)** |
| **2** | 3.45 (2.85–4.51)*** | 1.13 (1.03–3.00)* | 2.89 (2.04–7.76)* |
| **3** | 3.98 (3.42–5.00)*** | 2.31 (1.77–3.14)** | 3.54 (2.89–4.55)*** |
| **4** | 5.11 (4.17–6.57)*** | 3.56 (2.98–6.41)** | 4.72 (3.51–7.89)*** |
| **More than 4** | 7.03 (5.56–8.66)*** | 5.27 (4.09–9.15)*** | 5.01 (4.33–8.90)*** |
| **BMI** | | | |
| Underweight[R] | 1 | 1 | 1 |
| **Normal weight** | 1.27 (1.16–1.53)*** | 1.16 (0.82–1.61) | 0.78 (0.57–1.05) |
| **Overweight** | 2.27 (2.07–2.89)*** | 2.32 (1.58–3.40)*** | 1.12 (0.82–1.53) |
| **Obese** | 4.25 (3.29–5.51)*** | 2.51 (1.33–3.67)*** | 2.11 (1.51–2.93)*** |
| **Type of hospital** | | | |
| Public[R] | 1 | 1 | 1 |
| **Private** | 5.31 (4.76–5.93)*** | 2.03 (1.61–2.56)*** | 1.30 (1.08–1.54)*** |

Note: [R] stands for reference category and

*, **, and *** indicate significance at 5%, 1%, and 0.1% levels, respectively.

in the prevalence of C-sections by mothers' place of residence. In Bangladesh and Nepal, urban women are more inclined to undergo C-sections, but in Pakistan, urban women are less prone to C-sections. Previous researchers also found similar results [41, 45, 50, 51]. The reasons of lower rate of C-sections in rural areas in Bangladesh and Nepal could be due to a number of factors such as lack of accessibility to hospitals, customs and traditions of the society, financial ability, awareness, and others [52–56]. Availability of improved medical facilities, more antenatal care (ANC) visits, the tendency to take children at a later age, high socioeconomic status, highly educated mothers and their partners, increased BMI, and post-partum reasons may be some possible reasons for increased use of C-section in the urban regions. However, the lower number of C-sections among urban women in Pakistan warrants further investigation.

Consistent with several previous studies [23, 25, 40, 41, 45], we observed that women's (also husband's) educational attainment was associated with increased likelihood of choosing C-sections. Educated women were more predisposed to select C-section as a better method of delivery than the vaginal delivery. Maybe educated women feel that C-sections reduce delivery-related pain. Additionally, many educated women choose to take a child at a later stage of life compared to non-educated women, which increased their likelihood to choose the C-section methods. In the case of Nepal, women's education was not associated with C-section rates, which contradicts the previous studies [50, 51]. A further study may help to unravel the reasons. Birth order was reversely related to the prevalence of C-sections. A higher likelihood of C-sections was also evident for the first birth in all countries. Almost similar patterns were reported in the previous studies [39, 41, 46, 50]. Mothers in their second, third or more orders of birth may be less intimidated by the pain of labour, and they develop the physical and mental ability to cope with the delivery related pain. Although WHO suggested at least four antenatal care visits for every pregnant women to reduce pregnancy related complications [57], we found that C-section was associated with the increased number of antenatal care visits across the three countries. Maybe more antenatal care (ANC) influence anxious mothers to choose the C-section method, or physicians suggest them undertake C-section to relieve them from other pregnancy related complications [49]. The number of antenatal care (ANC) visits also interlinked with education and socioeconomic conditions. Several previous studies also in agreement with our findings regarding the association between number of antenatal care (ANC) visits and C-section [23, 25, 40, 41, 46, 51].

We found that there is an increased use of C-section delivery among women who possess the wealthiest economic status than those in economically less affluent positions in Bangladesh. This might happen due to the preference of many people with improved socioeconomic status to visit private hospitals for more comfortable healthcare than the public hospitals. C-section rate is noticeably higher in private hospital than public hospitals [55, 56]. With the increased affordability of women from the wealthiest economic status, doctors are likely to be in favour of not taking any risks and deciding on C-sections. Some other developing countries from Africa and South-Asia reported almost similar relationship between wealth quintile and C-section delivery [23, 24, 45, 58–60]. We found BMI is significantly associated with C-section delivery, with obese women were more exposed to C-section than the normal weight women. Several prior studies in different developing countries also exhibited the comparable and consistent results to our findings [23, 61–63]. The reasons may be much higher delivery-related complications for overweight and obese women than for normal weight women. Generally, overweight/obese women suffer from chronic diseases such as diabetes mellitus, asthma, hypertension, respiratory problems, and others. This may contribute to physicians' decisions for C-sections in reducing the risk.

## 5. Strengths and limitations

One key strength of our study is using representative data from the DHS for at least four rounds across the neighbouring countries. We conducted comprehensive analysis (both bivariate and multivariable) including a range of variables, which allowed us to understand how these variables are associated with C-section across the three countries. Our study also has some limitations. First, we could not establish the casual relationship as all DHSs are cross sectional in nature. In addition, there is a lack of valid instrumental variables (strongly associated with the independent variable, not dependent with the variable except only through an independent variable), which could be used to address simultaneity bias. Second, DHS data may be subject to recall bias and response bias since it is held in every three or five years in different countries; also, although survey weight was being used, the variability between the countries was not considered. Third, we fail to consider some important factors such as distance from residence to hospital, pregnancy related complexity due to unavailability in the DHS data sets. These factors may also mediate the association observed in our study. Fourth, as the study considered data from three countries, it may not be generalized to other South Asian countries. Finally, wealth condition is considered as a proxy for the socioeconomic status that is formed based on some household amenities that might underestimate or overestimate the actual socioeconomic condition. There is also no data related to the hospitals' inclination to be financially benefited by doing unnecessary caesarean delivery.

## 6. Conclusion

The rapid increase in C-sections across the three countries underscores that this procedure may not always be guided by clinical evidence in practice. Unwanted C-sections may potentially be an economic burden on the health care system as well as have detrimental health effects for both mothers and kids. Our findings regarding the association of sociodemographic factors with increased C-sections may help identify subgroups of women susceptible to C-sections and offer better support regarding C-sections plans. However, the substantial increase in C-sections across the three countries warrants further investigation to identify the reasons.

## Supporting information

**S1 Appendix. All the supplementary analysis and figures are provided in a separate file.**
This file consists of Table A1: Percentages of delivery by C-section in urban-rural places of residence in Bangladesh, Nepal, and Pakistan at different DHS rounds. Table A2: Percentages of C-section delivery by administrative division in Bangladesh, Nepal, and Pakistan at different DHS rounds. Figure A1: Trends of birth at hospitals for three South Asian Countries: Bangladesh, Nepal, and Pakistan. Figure A2: Trends of C-sections in the three South Asian Countries: Bangladesh, Nepal, and Pakistan.
(DOCX)

## Acknowledgments

We would like to acknowledge DHS for permitting in data use. Also, heartfelt gratitude to all the participants of the DHS of the three countries.

## Author Contributions

**Conceptualization:** Md Sohel Rana.

**Data curation:** Md Sohel Rana.

**Formal analysis:** Md Sohel Rana, Shrabanti Mazumder, Md Tareq Ferdous Khan, Md Mijanur Rahman.

**Investigation:** Md Mobarak Hossain Khan, Md Mijanur Rahman.

**Methodology:** Md Sohel Rana, Shrabanti Mazumder, Md Tareq Ferdous Khan, Md Mijanur Rahman.

**Software:** Md Sohel Rana, Md Tareq Ferdous Khan.

**Supervision:** Md Mijanur Rahman.

**Validation:** Md Sohel Rana, Shrabanti Mazumder, Md Tareq Ferdous Khan, Md Mijanur Rahman.

**Writing – original draft:** Md Sohel Rana, Md Mobarak Hossain Khan.

**Writing – review & editing:** Md Sohel Rana, Shrabanti Mazumder, Md Tareq Ferdous Khan, Md Mijanur Rahman.

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
