## [Decision Letter · Decision Letter 0]

27 Apr 2023

PONE-D-22-20214Trends and Determinants of Caesarean Section in South Asian Countries: Bangladesh, Nepal, and PakistanPLOS ONE

Dear Dr. Rana,

Thank you for submitting your manuscript to PLOS ONE. After careful consideration, we feel that it has merit but does not fully meet PLOS ONE’s publication criteria as it currently stands. Therefore, we invite you to submit a revised version of the manuscript that addresses the points raised during the review process.

We look forward to receiving your revised manuscript.

Kind regards,

Enamul Kabir

Academic Editor

PLOS ONE

2. Please upload a copy of Supporting Information Table S1 which you refer to in your text on page 8.

Reviewers' comments:

Reviewer's Responses to Questions

**Comments to the Author**

1. Is the manuscript technically sound, and do the data support the conclusions?

Reviewer #1: Yes

Reviewer #2: Partly

2. Has the statistical analysis been performed appropriately and rigorously? 

Reviewer #1: Yes

Reviewer #2: Yes

3. Have the authors made all data underlying the findings in their manuscript fully available?

Reviewer #1: Yes

Reviewer #2: Yes

4. Is the manuscript presented in an intelligible fashion and written in standard English?

Reviewer #1: Yes

Reviewer #2: Yes

5. Review Comments to the Author

Reviewer #1: 1. The author (s) must give proper justification to compare these three countries; saying these are neighbouring countries is not a sufficient argument. The justification/explanation must be from a healthcare/MCH perspective. Moreover, the Authors argued that the C-section rate in Nepal is still under the WHO recommended range, so there is no significance to comparing Nepal with Bangladesh and Pakistan.

2. The study's findings are similar to the DHS report; the author should go one step further to estimate the prevalence of caesarean section in rural/urban areas or public/private areas in these countries. Furthermore, it has been recommended that, author should estimate state / region-wise prevalence of caesarean section and construct GIS map for better visualization. It will also help to strengthen the current study and policymakers to identify the area with high c-section.

3. The author must include the variable "type of hospital/facility" in which Caesarean Section delivery have been performed in this analysis. It helps to understand why CS rises, especially because of private facilities across the counties. If these countries have any MCH/delivery subsidies scheme then need to explain in the discussion section.

4. Is Table 3 a pooled analysis of the association of c-sections with socio-demographic factors? If yes, need to write in the methodology section with justification and, if no, then mention the year in the table title. Similarly, for the odds ratio table 4, Is it pooled dataset?

5. The discussion section is weak; the author must discuss how private hospitals encourage c-section delivery for profit. Further, the study also discuss how unnecessary/higher caesarean delivery causes adverse health effects on the mother and child for a long time.

Reviewer #2: Reviewer Comments:

The authors address an important public health issue in the countries discussed. However, the manuscript does an unsatisfactory job of proving the premise, rationale and justification of the study goals. The manuscript does not seem well-written, (in many cases) overinterpret its results, and requires significant work to be of publishable quality.

Major points:

1.The manuscript provides considerably outdated, misappropriated and (in some cases completely irrelevant) references in the Background section. The authors were also unable to provide references where they are required even though the total number of references used is almost 60.

2.The study tends to over-interpret its results and does a very unsatisfactory job of describing the premise of the public health issue in the Background and conducting a literature review. The authors’ repeated use of similar analysis of the same data set from past years as the justification for some of their claims is problematic.

3.The authors’ rationale behind selecting the 3 countries was “Since Bangladesh, Nepal and Pakistan is neighbouring countries, so this study will enable to compare trends of C-section and its determinants”, which does not necessarily shed light behind exclusion of other South Asian countries (for example India shares border with all 3 countries and shares many of the same sociodemographic components but was not included in the analysis). Do the three countries selected have similar health infrastructure to facilitate direct comparisons? If so, in what ways are they similar? Do the socio-cultural dissimilarities prohibit/encourage elective C-sections for one country over the other?

4.In the Abstract the authors note that: variables which were significant in chi-square test included as explanatory variables. (There is no statistical basis for performing statistical analyses in this way). However, in the methods section it is stated that model selection was performed using “backward elimination” (which again is a flawed approach as model selection should be based on sociodemographic domain and theoretical considerations and not just statistical significance).

Abstract:

1Abstract: “There is a paucity of studies exploring trends and determinants of C-section in the three South-Asian countries: Bangladesh, Nepal and Pakistan”. This claim is not necessarily true as there have been studies on this issue in these countries. For example, the following papers are similar studies in Bangladesh and Nepal: doi: 10.1136/bmjopen-2014-005982, https://doi.org/10.1038/s41598-021-96337-0; also using same DHS (BD and Nepal) data from past years: doi.org/10.1371/journal.pone.0202879, https://doi.org/10.1186/s12884-020-03453-2, DOI: 10.1155/2021/8888267, and exact same data from Pakistan (https://doi.org/10.1186/s12884-020-03457-y).

2The manuscript does use the most recent DHS data, which is indeed correct. But these datasets have been released fairly recently.

3Although C-section is a lifesaving delivery method at emergency, unnecessary use of it is *a financial* burden for the family.

4“By reducing the influential factors respective authorities should minimize the over reliance on C-section during the childbirth” – Not sure how the authors could prescribe to reduce demographic attributes such as: mothers age, place of residence, years of education, partner’s years of education, wealth status, birth order, number of antenatal care visits, and body mass index.

5“Moreover, more mass media campaign should be initiated to create awareness among the stakeholders about the negative impacts of unnecessary C-sections” – The study did not consider media as a variable in the analyses and found any association. This is clearly extrapolating the results far outside the scope of the manuscript.

Background:

1.The first sentence in the Background talks about when C-sections are justified or not. However, this is not discussed further in the text. A better topic statement regarding the premise of the study could be considered instead of starting with the claim whether C-sections are justified or not.

2.The authors state: “During the past 30 years there was an increasing trends of practice of C-section, it varies from 0.4 to 40 percent among countries around the world [1]”. However, the study they cite is from 2006 and is based on data from 1993 to 2003. A correct citation is required.

3.The authors again state that: “In a recent study, which included data from 169 countries across the globe in 2015, the overall C-section rate was reported to be 21.1%, twice as high as that estimated based on 2000 [4]." However, the reference for this statement refers to a 2015 cross-sectional study in Tehran!

4.“Previously, it is observed that economically developed countries have a higher rate of C-section.” This requires a reference.

5.“The developed countries are strictly imposing restrictions on unnecessary C-sections.” – This requires a reference.

6.“There is a range of 0 to 10 percent of C-section among 76 percent of low-income countries, 16 percent of medium-income countries, and 3 percent of high-income countries. In addition, the distribution of having more than 20 percent of C-section is as: three percent in the low-income countries, 36 percent in medium income countries, and 31 percent in high-income countries [1].” The figures are again derived from a study performed on the data from 1991 – 2003. The authors need to report recent figures and provide the information in concise manner. The given text is difficult for a reader to follow.

7.The discussion pertaining to risks of C-sections is incoherent, lacks appropriate references (only studies from early 2000s are cited) and requires more work to be articulate. For example, the authors mention injury and disability as a risk of C-section. However, it is unclear from the text who is at risk, the mother or the baby or both.

8.The authors should note the revised statement regarding the recommended rate of C-sections (the commonly cited 10-15% range): “At population level, caesarean section rates higher than 10% are not associated with reductions in maternal and newborn mortality rates.” [link: https://www.who.int/publications/i/item/WHO-RHR-15.02]

9.Please fix the idiosyncrasies in the text: According to the WHO report 2010, there were 6 million elective C-section practiced performed in 2008 and caused cost which resulted in an estimated cost of….”. There are more of such inconsistencies.

10.“C-section in the private hospitals usually summed up higher medical care cost compared to government hospitals” – Please provide a reference from the countries discussed in the paper.

11.The authors state that, “Literatures found a number of determinants which are associated with the implementation of C-section including maternal age, maternal weight, parity, prolonged labour, mother with HIV infection, previous C-section delivery, dystocia, breech presentation, placenta previa, and suspected fatal complications [22-25]” –

a.The authors need to provide information on the class of determinants the existing literature discusses, rather than listing them as is currently provided.

b.The references here are from 1996, 1999, 2006 and 2007. The authors need to include up to date references.

There are other important studies the authors failed to discuss in the literature review. For example, the study claims that one of the main motivation behind the research is the rise in elective C-sections but fails to discuss the following papers that analyzes the determinants of Elective C-sections in Bangladesh (https://doi.org/10.1038/s41598-021-96337-0) and another relevant paper (https://doi.org/10.3390/ijerph19031465).

Methods:

1.The use of the phrase “waves of DHS” in this context is imprecise, please use the phrase "round of DHS". Also, the authors should refrain from using terms such as “cherished objectives”.

2.Section 2.1 is poorly worded. A brief discussion on study design should be included.

3.The references 30-32 are completely unnecessary (they are unrelated papers, and all are based on Bangladesh) and should be replaced with the DHS report books for each country.

4.The authors need to clarify what they mean by “Only the institutional deliveries held on governmental or non-governmental healthcare centres are considered as C-section deliveries.” Does this mean all institutional deliveries are C-sections? Which would not make sense.

5.“parity, number of ANC visits” need to be clearly defined. The acronym ANC does not have the full form.

6.“For model selection, backword elimination procedure was followed.” The model selection should be based on sociodemographic domain and theoretical considerations.

7.Instead of latest versions, please provide the version number for the statistical software used.

Results:

1.Information in tables 1 and 2 could be more concisely presented in a line graph with appropriate labelling, showing the trends for each country.

2.The authors should make sure they use the words enhanced and increased in a consistent manner.

3.The study is cross-section in nature; however, the authors use causal language to describe the associations. For example, the authors wrote, “For every one-year increase of women’s age, the risk of caesarean section is approximately 1.07, 1.11 and 1.02 times in women as compared to women with normal delivery in Bangladesh, Nepal and Pakistan respectively”. The correct form of interpretation should be, “For women who differ in age by 1 year but possess the same levels of other variables, the odds of having a C-section birth is approximately 1.07, 1.11 and 1.02 times the odds of having a normal/vaginal delivery in women from Bangladesh, Nepal and Pakistan respectively.”

4.Similar misinterpretation of Odds Ratio statistics can be seen in the form of “2.52 and 2.64 times more likely underwent caesarean delivery”. Likelihood is not equivalent to Odds ratios.

Discussion:

1.The manuscript says, “Along with the increase of delivery at facility the rate of C-section increases rapidly echoes unnecessary use of healthcare facilities” – This is a very strong statement to make from an observational study and without proper references. Furthermore, the response variable was a binary variable for having C-section or not. There is no further evidence from the study to support that most of the increase are due to “unnecessary use of healthcare facilities”. The reference cited (WHO study from 2006) only speaks to the second part of the claim “…. detrimental future effects for both mother and kids”.

2.“In Bangladesh the prevalence of C-section were more than two times in 2017” – Unclear statement. Two times regarding what year is unclear.

3.“Other studies findings also bolster our assertion [37, 38].” – This is redundant as both these studies use the DHS surveys used in the current manuscript and are of similar analyses.

4.“What are the reasons behind of such skyrocketing rates of C-section? Is only the improvement of medical science and healthcare facilities being the main cause?” - This is a very minor issue, but authors should refrain from asking direct questions to the readers, but rather pose them study inquiries.

5.“Most of the unnecessary C-section occurred due to mother’s misconception of delivery pain, convenience and lack of patience [43, 44]” – the references are from 2001 and 2003 studies based in the UK, which is demographically quite different from the countries discussed in the paper. The authors must remove these and include relevant and contemporary studies.

6.“Besides, physicians intention of earning more money is another significant factors of worldwide upsurge of elective C-section [45, 46]” – This again a very strong claim to make using references from 2003 and 1993, and may be outside the scope of the performed analyses without clarifying how the physicians stand to make more money since it is not mentioned whether physicians in Govt facilities in these countries are paid a specified salary or a variable salary based on the number of C-sections they perform.

7.“This association resonant with other studies [36, 38, 47, 48].” This is again a redundant claim since the first two studies used the exact same data (except maybe missing the most recent DHS).

8.“Premature and older women were adversely associated with reproductive and pregnancy related risks. Several studies divulged that elderly pregnant women underwent highly risks of preterm delivery, low birth weight, perinatal death, and CS [49, 50].” – Unclear context.

9.“In Bangladesh and Nepal urban women are more inclined to exploiting C-section, but in Pakistan rural women are more prone to C-section.” – Not sure how the authors came to such inference from statistically non-significant results (Table 4).

10.“increased BMI because of westernized food cultures may be the possible reasons of increased use of C-section” – This may be over-interpreting the scope of the analyses performed in the manuscript. BMI increase could be due to other factors such as post-partum issues. There is no indication in the data that the BMI is pre-pregnancy BMI.

11.“In Bangladesh and Pakistan, the likelihood of choosing C-section by pregnant women were associated with the educational attainment of both women and their husband matching with the results of other studies [27, 35, 36, 47].” - The authors cannot repeatedly claim other studies support their findings if those studies used the exact same data and analysis methods. For example, “In case of Nepal women education was not associated with the C-section rates, which contradicts with the previous studies [37, 38]” – this is an interesting result which requires more discussion with respect to the sociodemographic differences in Nepal compared to Bangladesh and Pakistan.

12.“… Similar pattern was documented in the previous studies [34, 36, 38, 48]” – Similar issue.

13.“Mothers during their second, third or more order of birth giving may be less intimidate the pain of labour as well as they develop the physical and mental ability to cope with the hazards of C-section.” – This is confusing. Are experienced mothers less intimidated by vaginal birth or C-sections? The study found age was positively associated with higher odds of C-section.

14.“Maybe more number of ANC influence anxious mothers to choose the C-section, or physicians inspire them to undertook C-section for getting more incentive from the hospitals.” – These claims need to be supported by evidence.

15.“It is expected that within 2026 Bangladesh will enter to the elite class of middle-income countries.” – The authors should provide references to such over-optimistic claims.

16.“As socioeconomic conditions have improved the ability of bearing healthcare costs of peoples also increased, that is why people prefer to go to the private hospitals for getting more comfortable healthcare than the public hospitals.” – Attributing people’s health seeking tendencies to economic development only is problematic. This again goes beyond the scope of the paper.

17.“Several prior studies in different developing countries in the world also exhibited the comparable and consistent results to our findings [27, 34, 55-59].” References 27, 34, 55 etc. are studies based in Bangladesh and Pakistan using same data. Please remove them.

Conclusion:

1.“As policy implications we are suggesting social media awareness for reducing unnecessary C-section.” This is the final concluding remark. However, the authors never really discussed why such methods would work in the context of this specific public health problem.

6. PLOS authors have the option to publish the peer review history of their article (what does this mean?). If published, this will include your full peer review and any attached files.

Reviewer #1: No

Reviewer #2: No

---

## [Author Response · Author response to Decision Letter 0]

25 Jul 2023

Please have a look at the attached 'response to reviewers' document.

---

## [Decision Letter · Decision Letter 1]

24 Aug 2023

PONE-D-22-20214R1Trends and Determinants of Caesarean Section in South Asian Countries: Bangladesh, Nepal, and PakistanPLOS ONE

Dear Dr. Rana,

Thank you for submitting your manuscript to PLOS ONE. After careful consideration, we feel that it has merit but does not fully meet PLOS ONE’s publication criteria as it currently stands. Therefore, we invite you to submit a revised version of the manuscript that addresses the points raised during the review process.

We look forward to receiving your revised manuscript.

Kind regards,

Enamul Kabir

Academic Editor

PLOS ONE

Reviewers' comments:

Reviewer's Responses to Questions

**Comments to the Author**

1. If the authors have adequately addressed your comments raised in a previous round of review and you feel that this manuscript is now acceptable for publication, you may indicate that here to bypass the “Comments to the Author” section, enter your conflict of interest statement in the “Confidential to Editor” section, and submit your "Accept" recommendation.

Reviewer #1: All comments have been addressed

Reviewer #2: (No Response)

2. Is the manuscript technically sound, and do the data support the conclusions?

Reviewer #1: Yes

Reviewer #2: Partly

3. Has the statistical analysis been performed appropriately and rigorously? 

Reviewer #1: Yes

Reviewer #2: Yes

4. Have the authors made all data underlying the findings in their manuscript fully available?

Reviewer #1: Yes

Reviewer #2: Yes

5. Is the manuscript presented in an intelligible fashion and written in standard English?

Reviewer #1: Yes

Reviewer #2: Yes

6. Review Comments to the Author

Reviewer #1: The author has successfully addressed the previous comments, resulting in a significant improvement in the manuscript's quality.

Reviewer #2: The manuscript has been worked on by the author since the initial review. However, it is still Well-below the quality and intelligibility required for publication.

Summary:

Based on the potential merits and existing demerits of the paper, I would suggest either a Rejection or another Major Revision where authors should contextualize the findings of their study, include the suggested variable into the analysis, and fix all the issues that they did not fix in the first revision.

In summary, aside from the fact that the data is novel and there are a few interesting results (which the authors again were not able to discuss properly in the revision), the manuscript is nowhere near publishable quality and is still very poorly written.

Major issues still existing:

1. The authors still claim in the abstract and in the manuscript that “Through modification of influential factors (body mass index, birth order, wealth, and education) respective authorities should minimize the over reliance on C-section during childbirth.”

- For example, how can “authorities” influence “birth order”, given that the authors show in their analyses that “there is an 88% less chance of caesarean delivery among the women who are giving more than four birth than the women who are giving the first birth)”?

Do they mean they want the authorities to make sure that women avoid their first few births to lessen the odds of a C-section? How does that even make sense? Or do they want only older women with many children to give birth in the future? This may sound harsh but the authors, I think, do not really think about the implications of their recommendations. What they can really argue to be at play is that younger mothers tend to show a tendency to opt for a C-section, which may be portrayed to them as an easier/lower-risk/lower-painful option under anesthesia compared to normal vaginal birth. This is just one example of incoherent discussion and interpretation by the authors.

- Again, how can modifying education level help reduce C-section? The authors show in their analyses that higher the education level, the higher odds of observing C-section. Do they mean they would like the authority to curtail women’s education to keep C-section rates low? This does Not make sense.

2. The authors finally argue their rationale for selecting the 3 countries that they chose based on the fact that “they have several similarities in terms of maternal and child health: high maternal and child mortality rates, limited access to quality health care, malnutrition and undernutrition, maternal health services, socio-economic and cultural factors” as well as similar challenges. However, at the same time they highlight that Nepal is different from the other two countries later in the paper and this is also apparent from their results. Furthermore, they do not provide any evidence/reference to the reader for the above argument for their study rationale.

3. In response to the comment “The author must include the variable "type of hospital/facility" in which Caesarean Section delivery have been performed in this analysis. It helps to understand why CS rises, especially because of private facilities across the counties”, the authors do not acknowledge its importance and shy away by saying that they will incorporate this into their future study. Which makes no sense as whether the current manuscript will be judged on what is included in this one and not what the authors promise to do for their next study.

4. One of the main arguments of their study is the “WHO recommended standard level of C-section (10%-15%)”. However, this figure has been widely criticized and even the WHO no longer supports this figure in the context of the current scenario across the world.

5. The authors continue to use inconsistent and irrelevant references to support strong claims. One example is: “Besides, physicians intention of earning more money is another significant factors of worldwide upsurge of elective C-section [39]” This reference comes from a Swedish study, which is not relevant to LMICs. Even the authors do not argue how a study on economics of C-section in a developed country can be used to justify a claim in the context of the entire world.

6. The study still misses the mark of its intended target. For example, “Our study also depicted that there is inequality of the prevalence of C-section by place of residence of the mothers. In Bangladesh and Nepal, urban women are more inclined to undergo C-section, but in Pakistan, urban women are less prone to C-section.” The authors do not go into discussing why this might be case or why the difference exists for rural women of these three countries. They provide convoluted sentence a bit later trying to apparently justify all the interesting results at once, which again makes little sense. The paper is about understanding the dynamics, similarities, and dissimilarities of the three discussed countries that “share a lot of the same demographic characteristics”.

7. The authors made little attempt at fixing the Discussion section with one or two minor changes, which was one of the most problematic areas of the original manuscript.

8. The strengths and limitations discussed are also very generic and do not speak to the issues unique to this study.

9. The manuscript still offers very little insight into the possible dynamics of the trends and determinants, which was their main goal of the study.

Minor points:

There are still quite a few grammatical issues and these need to be fixed.

If possible, the authors can also consult a public healthcare/medical professional regarding the findings of their analyses to put their results into more context so that it can be helpful to policy discussions in the future.

7. PLOS authors have the option to publish the peer review history of their article (what does this mean?). If published, this will include your full peer review and any attached files.

Reviewer #1: No

Reviewer #2: No

---

## [Author Response · Author response to Decision Letter 1]

7 May 2024

Response to reviewer comments are attached in document.

---

## [Decision Letter · Decision Letter 2]

11 Jul 2024

PONE-D-22-20214R2Trends and Determinants of Caesarean Section in South Asian Countries: Bangladesh, Nepal, and PakistanPLOS ONE

Dear Dr. Rana,

Thank you for submitting your manuscript to PLOS ONE. After careful consideration, we feel that it has merit but does not fully meet PLOS ONE’s publication criteria as it currently stands. Therefore, we invite you to submit a revised version of the manuscript that addresses the points raised during the review process.

We look forward to receiving your revised manuscript.

Kind regards,

Enamul Kabir

Academic Editor

PLOS ONE

Journal Requirements:

Reviewers' comments:

Reviewer's Responses to Questions

**Comments to the Author**

1. If the authors have adequately addressed your comments raised in a previous round of review and you feel that this manuscript is now acceptable for publication, you may indicate that here to bypass the “Comments to the Author” section, enter your conflict of interest statement in the “Confidential to Editor” section, and submit your "Accept" recommendation.

Reviewer #1: All comments have been addressed

Reviewer #2: All comments have been addressed

2. Is the manuscript technically sound, and do the data support the conclusions?

Reviewer #1: Yes

Reviewer #2: Yes

3. Has the statistical analysis been performed appropriately and rigorously? 

Reviewer #1: Yes

Reviewer #2: Yes

4. Have the authors made all data underlying the findings in their manuscript fully available?

Reviewer #1: Yes

Reviewer #2: Yes

5. Is the manuscript presented in an intelligible fashion and written in standard English?

Reviewer #1: Yes

Reviewer #2: Yes

6. Review Comments to the Author

Reviewer #1: The author has revised the manuscript, making it more presentable. The findings of the study will highlight the progress of cesarean births in South Asian countries.

Reviewer #2: The current version of the manuscript is substantially improved and well-written. I thank the authors for their patience and effort to address all my concerns. I believe the manuscript can now be accepted in its current form after the following minor comments have been addressed.

Minor comments:

1. The figures A1 and A2 should be fixed in the following manner:

a. Both figures should the same same X-axis. A1 has x = 0, while A2 does not. Furthermore, the axis ticks and marks also differ for the two plots. They should be the same and have the same starting point.

b. There should be uniform color for the same country in the two plots. For example, Bangladesh has blue and black trendlines in the two plots. It should be the same color. The color scheme for Nepal is a good example.

2. The keyword "Cesarean" in the submission portal information (first page of the generated pdf) spells differently from what is followed in the manuscript (Caesarean). Please take note and maybe fix it in the final version.

7. PLOS authors have the option to publish the peer review history of their article (what does this mean?). If published, this will include your full peer review and any attached files.

Reviewer #1: No

Reviewer #2: No

---

## [Author Response · Author response to Decision Letter 2]

12 Jul 2024

Thanks a lot for your valuable inputs. Your criticisms, suggestions helps us to improve the quality of our work. Please have a look at the response to reviewers file.

---

## [Decision Letter · Decision Letter 3]

12 Sep 2024

Trends and Determinants of Caesarean Section in South Asian Countries: Bangladesh, Nepal, and Pakistan

PONE-D-22-20214R3

Dear Dr. Rana,

We’re pleased to inform you that your manuscript has been judged scientifically suitable for publication and will be formally accepted for publication once it meets all outstanding technical requirements including the need for English language grammar and style editing. 

Kind regards,

Calistus Wilunda, DrPH

Academic Editor

PLOS ONE

Additional Editor Comments (optional):

 The manuscript needs English language grammar and style editing before formal acceptance. 

Reviewers' comments:

Reviewer's Responses to Questions

**Comments to the Author**

1. If the authors have adequately addressed your comments raised in a previous round of review and you feel that this manuscript is now acceptable for publication, you may indicate that here to bypass the “Comments to the Author” section, enter your conflict of interest statement in the “Confidential to Editor” section, and submit your "Accept" recommendation.

Reviewer #1: All comments have been addressed

Reviewer #2: All comments have been addressed

2. Is the manuscript technically sound, and do the data support the conclusions?

Reviewer #1: Partly

Reviewer #2: Yes

3. Has the statistical analysis been performed appropriately and rigorously? 

Reviewer #1: Yes

Reviewer #2: Yes

4. Have the authors made all data underlying the findings in their manuscript fully available?

Reviewer #1: Yes

Reviewer #2: Yes

5. Is the manuscript presented in an intelligible fashion and written in standard English?

Reviewer #1: Yes

Reviewer #2: Yes

6. Review Comments to the Author

Reviewer #1: (No Response)

Reviewer #2: The current version of the manuscript is well written. I again thank the authors for their patience and effort to

address my concerns. I believe the manuscript can now be accepted in its current form.

7. PLOS authors have the option to publish the peer review history of their article (what does this mean?). If published, this will include your full peer review and any attached files.

Reviewer #1: No

Reviewer #2: No

---

## [Editor Report · Acceptance letter]

30 Sep 2024

PONE-D-22-20214R3 

PLOS ONE

Dear Dr. Rana, 

I'm pleased to inform you that your manuscript has been deemed suitable for publication in PLOS ONE. Congratulations! Your manuscript is now being handed over to our production team.

Kind regards, 

on behalf of

Dr. Calistus Wilunda 

Academic Editor

PLOS ONE